# Optimizing Energy Production Using Policy Search and Predictive State Representations

**Yuri Grinberg**      **Doina Precup**
School of Computer Science, McGill University
Montreal, QC, Canada
{ygrinb,dprecup}@cs.mcgill.ca

**Michel Gendreau**∗
École Polytechnique de Montréal
Montreal, QC, Canada
michel.gendreau@cirrelt.ca

## Abstract

We consider the challenging practical problem of optimizing the power production of a complex of hydroelectric power plants, which involves control over three continuous action variables, uncertainty in the amount of water inflows and a variety of constraints that need to be satisfied. We propose a policy-search-based approach coupled with predictive modelling to address this problem. This approach has some key advantages compared to other alternatives, such as dynamic programming: the policy representation and search algorithm can conveniently incorporate domain knowledge; the resulting policies are easy to interpret, and the algorithm is naturally parallelizable. Our algorithm obtains a policy which outperforms the solution found by dynamic programming both quantitatively and qualitatively.

## 1   Introduction

The efficient harnessing of renewable energy has become paramount in an era characterized by decreasing natural resources and increasing pollution. While some efforts are aimed towards the development of new technologies for energy production, it is equally important to maximize the efficiency of existing sustainable energy production methods [5], such as hydroelectric power plants. In this paper, we consider an instance of this problem, specifically the optimization of one of a complex of hydroelectric power plants operated by Hydro-Québec, the largest hydroelectricity producer in Canada [17].

The problem of optimizing hydroelectric power plants, also known as the reservoir management problem, has been extensively studied for several decades and a variety of computational methods have been applied to solve it (see e.g. [3, 4] a for literature review). The most common approach is based on dynamic programming (DP) [13]. However, one of the major obstacles of this approach lies in the difficulty of incorporating different forms of domain knowledge, which are key to obtaining solutions that are practically relevant. For example, the optimization is subject to constraints on water levels which might span several time-steps, making them difficult to integrate into typical DP-based algorithms. Moreover, human decision makers in charge of the power plants are reluctant to rely on black-box closed loop policies that are hard to understand. This has led to continued use in the industry of deterministic optimization methods that provide long-term open loop policies; such policies are then further adjusted by experts [2]. Finally, despite the different measures taken to relieve the curse of dimensionality in DP-style approaches, it remains a big concern for large scale problems.

In this paper, we develop and evaluate a variation of simulation–based optimization [16], a special case of *policy search* [6], which combines some aspects of stochastic gradient descent and block

---

∗NSERC/Hydro-Québec Industrial Research Chair on the Stochastic Optimization of Electricity Generation, CIRRELT and Département de Mathématiques et de Génie Industriel, École Polytechnique de Montréal.

coordinate descent [14]. We compare our solution to a DP-based solution developed by Hydro-Québec based on historical inflow data, and show both quantitative and qualitative improvement. We demonstrate how domain knowledge can be naturally incorporated into an easy-to-interpret policy representation, as well as used to guide the policy search algorithm. We use a type of predictive state representations [9, 10] to learn a model for the water inflows. The policy representation further leverages the future inflow predictions obtained from this model. The approach is very easy to parallelize, and therefore easily scalable to larger problems, due to the availability of low-cost computing resources. Although much effort in this paper goes to analyzing and solving one specific problem, the proposed approach is general and could be applied to any sequential optimization problems involving constraints. At the end of the paper, we summarize the utility of this approach from a domain–independent perspective.

The paper is organized as follows. Sec. 2 provides information about the hydroelectric power plant complex (needed to implement the simulator used in the policy search procedure) and describes the generative model used by Hydro-Québec to generate inflow data with similar statistical properties as inflows observed historically. Sec. 3 describes the learning algorithm that produces a predictive model for the inflows, based on recent advances in predictive state representations. In Sec. 4 we present the policy representation and the search algorithm. Sec. 5 presents a quantitative and qualitative analysis of the results, and Sec. 6 concludes the paper.

## 2   Problem specification

We consider a hydroelectric power plant system consisting of four sites, $R_1, \ldots, R_4$ operating on the same course of water. Although each site has a group of turbines, we treat this group as a single large turbine whose speed is to be controlled. $R_4$ is the topmost site, and water turbined at reservoir $R_i$ flows to $R_{i-1}$ (where it gets added to any other naturally incoming flows). The topmost three sites ($R_2, R_3, R_4$) have their own reservoirs, in which water accumulates before being pushed through a number of turbines which generate the electricity. However, some amount of water might not be useful for producing electricity because it is *spilled* (e.g., to prevent reservoir overflow). Typically, policies that manage to reduce spillage produce more power.

The amount of water in each reservoir changes as a function of the water turbined/spilled from the upstream site, the water inflow coming from the ground, and the amount of water turbined/spilled at the current site, as follows:
$$V_4(t+1) = V_4(t) + I_4(t) - X_4(t) - Y_4(t),$$
$$V_i(t+1) = V_i(t) + X_{i+1}(t) + Y_{i+1}(t) + I_i(t) - X_i(t) - Y_i(t), i = 2, 3$$
where $V_i(t)$ is the volume of water at reservoir $R_i$ at time $t$, $X_i(t)$ is the amount of water turbined at $R_i$ at time $t$, $Y_i(t)$ is the amount of water spilled at site $R_i$ at time $t$, and $I_i(t)$ is water inflow to site $R_i$ at time $t$. Since $R_1$ does not have a reservoir, all the incoming water is used to operate the turbine, and the extra water is spilled. At the other sites, the water spillage mechanism is used only as a means to prevent reservoir overflow.

The control problem that needs to be solved is to determine the amount of water to turbine during each period $t$, in order to maximize power production, while also satisfying constraints on the water level. We are interested in a problem considered of intermediate temporal resolution, in which a control action at each of the 3 topmost sites is chosen weekly, after observing the state of the reservoirs and the inflows of the previous week.

**Power production model**

The amount of power produced is a function of the current water level (headwater) at the reservoir and the total speed of the turbines ($m^3/s$). It is not a linear function, but it is well approximated by a piece-wise linear function for a fixed value of the headwater (see Fig. A.1 in the supplementary material) . The following equation is used to obtain the power production curve for other values of the headwater [18]:
$$P(x, h) = \left( \frac{h}{h_{ref}} \right)^{1.5} \cdot P_{ref} \left( \left[ \frac{h}{h_{ref}} \right]^{-0.5} \cdot x \right), \tag{1}$$
where $x$ is the flow, $h$ is the current headwater level, $h_{ref}$ is the reference headwater, and $P_{ref}$ is the production curve of the reference headwater. Note that Eq. 1 implies that the maximum total

speed of the turbines also changes as the headwater changes; specifically, $\left[\frac{h}{h_{ref}}\right]^{-0.5} x$ should not exceed the maximum total speed of the turbines, given in the appendix figures. For completeness, Figure A.2 (supplementary material) can be used to convert the amount of water in the reservoir to the headwater value.

**Constraints**

Several constraints must be satisfied while operating the plant, which are ecological in nature.

1. Minimum turbine speed at $R_1$ ($MIN\_FLOW(w), w \in \{1, ..., 52\}$):

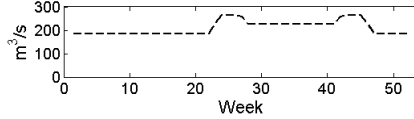

   This sufficient flow needs to be maintained to allow for easy passage for the fish living in the river.

2. Stable turbine speed throughout weeks 43-45 (fluctuations of up to $BUFFER = 35\ m^3/s$ between weeks are acceptable). Nearly constant water flow at this time of the year ensures that the area is favorable for fish spawning.

3. The amount of water in reservoir $R_2$ should not go below $MIN\_VOL = 1360\ hm^3$. Due to the depth of the reservoir, the top and bottom water temperatures differ. Turbining warmer water (at reservoir's top) is preferrable for the fish, but this constraint is less important than the previous two.

**Water inflow process**

The operation of the hydroelectric power plant is almost entirely dependent on the inflows at each site. Historical data suggests that it is safe to assume that the inflows at different sites in the same period $t$ are just scaled values of each other. However, there is relatively little data available to optimize the problem through simulation: there are only 54 years of inflow data, which translates into 2808 values (one value per week - see Fig. 1). Hydro-Quebec use this data to learn a generative model for inflows. It is a periodic autoregressive model of first order, PAR(1), whose structure is well aligned with the hydrological description of the inflows [1]. The model generates data using the following equation:

$$x(t + 1) = \alpha_{t \bmod N} \cdot x(t) + \xi(t),$$

where $\xi(t) \sim \mathcal{N}(0, \nu_{t \bmod N})$ i.i.d., $x(0) = \xi(0)$, and $N = 52$ in our setting.
As the weekly historical data is not necessarily normally distributed, transformations are used to normalize the data before learning the parameters of the PAR(1) model. The transformations used here are either logarithmic, $ln(X + a)$, where $a$ is a parameter, or gamma, based on Wilson Hilferty transformation [15]. Hence, to generate synthetic data, the reverse of these transformations are applied to the output produced by the PAR(1) process[1].

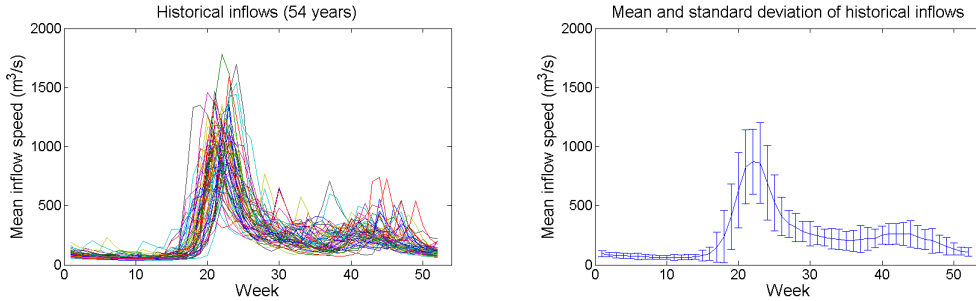

Figure 1: Historical inflow data.

[1]The parameters of the PAR(1) process, as well as the transformations and their parameters (in the logarithmic case) are estimated using the SAMS software [11].

# 3 Predictive modeling of the inflows

It is intuitively clear that predicting future inflows well could lead to better control policies. In this section, we describe the model that lets us compute the predictions of future inflows, which are used as an input to policies. We use a recently developed time series modelling framework based on predictive state representations (PSRs) [9, 10], called mixed-observable PSRs (MO-PSR) [8]. Although one could estimate future inflows based on knowledge that the generative process is PAR(1), our objective is to use a general modelling tool that does not rely on this assumption, for two reasons. First, decoupling the generative model from the predictive model allows us to replace the current generative model with more complex alternatives later on, with little effort. Moreover, more complex models do not necessary have a clear way to estimate a sufficient statistic from a given history (see e.g. temporal disaggregation models [12]). Second, we want to test the ability of predictive state representations, which are a fairly recent approach, to produce a model that is useful in a real-world control problem. We now describe the models and learning algorithms used.

## 3.1 Predictive state representations

(Linear) PSRs were introduced as a means to represent a partially observable environment without explicitly modelling latent states, with the goal of developing efficient learning algorithms [9, 10]. A predictive representation is only required to keep some form of sufficient statistic of the past, which is used to predict the probability of future sequences of observations generated by the underlying stochastic process.

Let $\mathcal{O}$ be a discrete observation space. With probability $\mathrm{P}(o_1, ..., o_k)$, the process outputs a sequence of observations $o_1, ..., o_k \in \mathcal{O}$. Then, for some $n \in \mathbb{N}$, the set of parameters

$$\{\mathbf{m}_* \in \mathbb{R}^n, \{\mathbf{M}_o \in \mathbb{R}^{n \times n}\}_{o \in \mathcal{O}}, \mathbf{p}_0 \in \mathbb{R}^n\}$$

defines a $n$-dimensional linear PSR that represents this process if the following holds:

$$\forall k \in \mathbb{N}, o_i \in \mathcal{O} : \mathrm{P}(o_1, ..., o_k) = \mathbf{m}_*^\top \mathbf{M}_{o_k} \cdots \mathbf{M}_{o_1} \mathbf{p}_0,$$

where $\mathbf{p}_0$ is the initial state of the PSR [7]. Let $\mathbf{p}(h)$ be the PSR state corresponding to a history $h$. Then, for any $o \in \mathcal{O}$, it is possible to track a sufficient statistic of the history, which can be used to make any future predictions, using the equation:

$$\mathbf{p}(ho) \triangleq \frac{\mathbf{M}_o \mathbf{p}(h)}{\mathbf{m}_*^\top \mathbf{M}_o \mathbf{p}(h)}.$$

Because PSRs are very general, learning can be difficult without exploiting some structure of the problem domain. In our problem, knowing the week of the year gives significant information to the predictive model, but the model does not need to learn the dynamics of this variable. This turns out to be a special case of the so-called *mixed observable PSR* model [8], in which an observation variable can be used to decompose the problem into several, typically much smaller, problems.

## 3.2 Mixed-observable PSR for inflow process

We define the discrete observation space $\mathcal{O}$ by discretizing the space of inflows into 20 bins, then follow [8] to estimate a MO-PSR representation from $3 \times 10^5$ trajectories obtained from the generative model. This procedure is a generalization of the spectral learning algorithm developed for PSRs [7], which is a consistent estimator.

Specifically, let the set of all observed tuples of sequences of length 3 be denoted by $\mathcal{H}$ and $\mathcal{T}$ simultaneously. We then split the set $\mathcal{H}$ into 52 subsets, each corresponding to a different week of the year, and obtain a collection $\{\mathcal{H}_w\}_{w \in \mathcal{W}}$, where $\mathcal{W} = \{1, ..., 52\}$. Then, we estimate a collection of the following vectors and matrices from data:

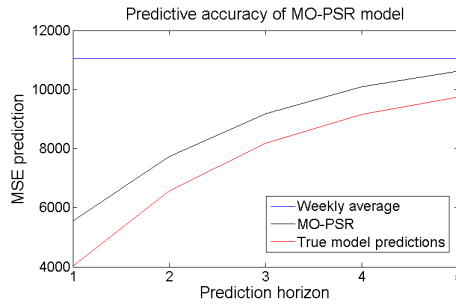

Figure 2: Prediction accuracy of the mean predictor (blue), MO-PSR predictor (black), and the predictions calculated from a true model (red).

- $\{\mathbf{P}_{\mathcal{H}_w}\}_{w \in \mathcal{W}}$ - a set of $|\mathcal{H}_w|$-dimensional vectors with entries equal to $\mathrm{P}(h \in \mathcal{H}_w|h$ occured right before week $w)$,

- $\{\mathbf{P}_{\mathcal{T},\mathcal{H}_w}\}_{w \in \mathcal{W}}$ - a set of $|\mathcal{T}| \times |\mathcal{H}_w|$-dimensional matrices with entries equal to $\mathrm{P}(h,t|h \in \mathcal{H}_w, t \in \mathcal{T}, h$ occured right before week $w)$,

- $\{\mathbf{P}_{\mathcal{T},o,\mathcal{H}_w}\}_{w \in \mathcal{W}, o \in \mathcal{O}}$ - a set of $|\mathcal{T}| \times |\mathcal{H}_w|$-dimensional matrices with entries equal to $\mathrm{P}(h,o,t|h \in \mathcal{H}_w, o \in \mathcal{O}, t \in \mathcal{T}, h$ occured right before week $w)$.

Finally, we perform Singular Value Decomposition (SVD) on the estimated matrices $\{\mathbf{P}_{\mathcal{T},\mathcal{H}_w}\}_{w \in \mathcal{W}}$ and use their corresponding low rank matrices of left singular vectors $\{\mathbf{U}_w\}_{w \in \mathcal{W}}$ to compute the MO-PSR parameters as follows:

- $\forall o \in \mathcal{O}, w \in \mathcal{W} : \mathbf{B}_o^w = \mathbf{U}_{w-1}^\top \mathbf{P}_{\mathcal{T},o,\mathcal{H}_w} (\mathbf{U}_w^\top \mathbf{P}_{\mathcal{T},\mathcal{H}_w})^\dagger,$

- $\forall w \in \mathcal{W} : \mathbf{b}_0^w = \mathbf{U}_w^\top \mathbf{P}_{\mathcal{T},\mathcal{H}_w} \mathbf{1},$

- $\forall w \in \mathcal{W} : \mathbf{b}_*^w = (\mathbf{P}_{\mathcal{T},\mathcal{H}_w}^\top \mathbf{U}_w)^\dagger \mathbf{P}_{\mathcal{H}_w},$

where $w - 1$ is the week before $w$, and $\dagger$ denotes the Moore–Penrose pseudoinverse. The above parameters can be used to estimate probability of any sequence of future observations, given starting week $w$, as:

$$\mathrm{P}(o_1, ..., o_t) = \mathbf{b}_*^{w+t\top} \mathbf{B}_{o_t}^{w+t-1} \cdots \mathbf{B}_{o_1}^w \mathbf{b}_0^w,$$

where $w + i$ represents the $i$-th week after $w$.

Figure 2 shows the prediction accuracy of the learnt MO-PSR model at different horizons, compared to two baselines: the weekly average, and the true PAR(1) model that knows the hidden state (oracle predictor).

## 4 Policy search

The objective is to maximize the expected return, $\mathrm{E}(R)$, over each year, given by the amount of power produced that year minus the penalty for constraint violations. Specifically,

$$R = \sum_{w=1}^{52} \left[ P(w) - \sum_{i=1}^{3} \alpha_i C_i(w) \right],$$

where $P(w)$ is the amount of power produced during week $w$, and $C_i(w)$ is the penalty for violating the $i$-th constraint, defined as:

$$C_1(w) = \min\{MIN\_FLOW(w) - R_1 flow(w), 0\}^2$$

$$C_2(w) = \begin{cases} \min\{|R_1 flow(w) - mean R_1 flow| - BUFFER, 0\}^2 & \text{if } w \in \{43, 44, 45\} \\ 0 & \text{otherwise} \end{cases}$$

$$C_3(w) = \min\{MIN\_VOL - R_2 vol(w), 0\}^{3/2}$$

where $R_1 flow(w)$ is the water flow (turbined + spilled) at $R_1$ during week $w$, $R_2 vol(w)$ is the water volume at $R_2$ at the end of week $w$, and $mean R_1 flow$ is the average water flow at site $R_1$ during weeks 43-45. There are three variables to control: the speed of turbines $R_2, R_3, R_4$. As discussed, the speed of the turbine at site $R_1$ is entirely controlled by the amount of incoming water.

The approach we take belongs to a general class of policy search methods [6]. This technique is based on the ability to simulate policies, and the algorithm will typically output the policy that has achieved the highest reward during the simulation.

The policy for each turbine takes the parametric form of a truncated linear combination of features:

$$\min \left[ \max \left( \sum_{i=1}^{k} x_j \cdot \theta_j, MAX\_SPEED_{R_i} \right), 0 \right],$$

where $MAX\_SPEED_{R_i}$ is the maximum speed of the turbine at $R_i$, $x_j$ are the features and $\theta_j$ are the parameters. For each site, the features include the current amount of water in the reservoir, the total amount of water in downstream reservoirs, and a constant. For the policy that uses the predictive

model we include one more feature per site: the expected amount of inflow for the following week. Hence, there are 8 and 11 features for the policies without/with predictions respectively (as there are no downstream reservoirs for $R_2$).

Using this policy representation results in reasonable performance, but a closer look at constraint 2 during simulation reveals that the reservoirs should not be too full; otherwise, there is a high chance of spillage, preventing the ability to set a stable flow during the three consecutive weeks critical for fish spawning. To address this concern, we use a different set of parameters during weeks 41-43, to ensure that the desired state of the reservoirs is reached before the constrained period sets in. Note that the policy search framework allows us to make such an adjustment very easily.

Finally, we also use the structure of the policy to comply as much as possible with constraint 2, by setting the speed of the turbine at site $R_2$ during weeks 44-45 to be equal to the previous water flow at site $R_1$. For the policy that uses the predictive model, we further refine this by subtracting the expected predicted amount of inflow at site $R_1$. This brings the number of parameters used for the policies to 16 and 22 respectively. As the policies are simply (truncated) linear combinations of features, they are easy to inspect and interpret.

Our algorithm is based on a random local search around the current solution, by perturbing different blocks of parameters while keeping others fixed, as in block coordinate descent [14]. Each time a significantly better solution than the current one is found, line search is performed in the direction of improvement. The pseudo-code is shown in Alg. 1. The algorithm itself, like the policy representation, exploits problem structure by also searching the parameters of a single turbine as part of the overall search procedure.

---

**Algorithm 1** Policy search algorithm

**Parameters:**
$N-$ maximum number of interations
$\boldsymbol{\theta} = \{\boldsymbol{\theta}_{R_2}, \boldsymbol{\theta}_{R_3}, \boldsymbol{\theta}_{R_4}\} = \{\theta_1, ..., \theta_m\} \in \mathbb{R}^m$ - initial parameter vector
$n-$ number of parallel policy evaluations
$Threshold-$ significance threshold
$\gamma-$ sampling variance
**Output:** $\boldsymbol{\theta}$

  1: **repeat**
  2:     **Stage 1:**                                         ▷ searching over entire parameter space
  3:         $\boldsymbol{\theta} = $ SEARCHWITHINBLOCK$(\boldsymbol{\theta}$, all indexes$)$
  4:     **Stage 2:**                       ▷ searching over parameters of each turbine separately
  5:         **for all** reservoirs $R_j$ **do**
  6:             $\boldsymbol{\theta} = $ SEARCHWITHINBLOCK$(\boldsymbol{\theta}$, parameter indexes of turbine $R_j)$
  7:     **Stage 3:**                             ▷ searching over each parameter separately
  8:         **for** $j \leftarrow 1, m$ **do**
  9:             $\boldsymbol{\theta} = $ SEARCHWITHINBLOCK$(\boldsymbol{\theta}$, index $j)$
10: **until** no improvement at any stage
11:
12: **procedure** SEARCHWITHINBLOCK$(\boldsymbol{\theta}, \mathcal{I})$          ▷ $\mathcal{I}, \mathcal{I}^c$ - an index set and its complement
13:     **repeat**
14:         Obtain $n$ samples $\{\Delta_i \sim \mathcal{N}(0, \gamma\mathbf{I})\}_{i\in\{1,...,n\}}$
15:         Evaluate policies defined by parameters $\{\{\boldsymbol{\theta}_{\mathcal{I}^c}, \boldsymbol{\theta}_{\mathcal{I}} + \Delta_i\}\}_{i\in\{1,...,n\}}$ *(in parallel)*
16:         **if** $\hat{E}(R_{\{\boldsymbol{\theta}_{\mathcal{I}^c}, \boldsymbol{\theta}_{\mathcal{I}} + \Delta_i\}}) > \hat{E}(R_{\boldsymbol{\theta}}) + Threshold$ **then**
17:             Find $\alpha_* = \arg\max_\alpha \hat{E}(R_{\{\boldsymbol{\theta}_{\mathcal{I}^c}, \boldsymbol{\theta}_{\mathcal{I}} + \alpha\Delta_i\}})$ using a line search
18:             $\boldsymbol{\theta} \leftarrow \{\boldsymbol{\theta}_{\mathcal{I}^c}, \boldsymbol{\theta}_{\mathcal{I}} + \alpha_*\Delta_i\}$
19:     **until** no improvement for $N$ consecutive iterations
20:     **return** $\boldsymbol{\theta}$

---

The estimate of the expected reward of a policy is calculated by running the simulator on a single 2000-year-long trajectory obtained from the generative model described in Sec. 2. Since the algo-

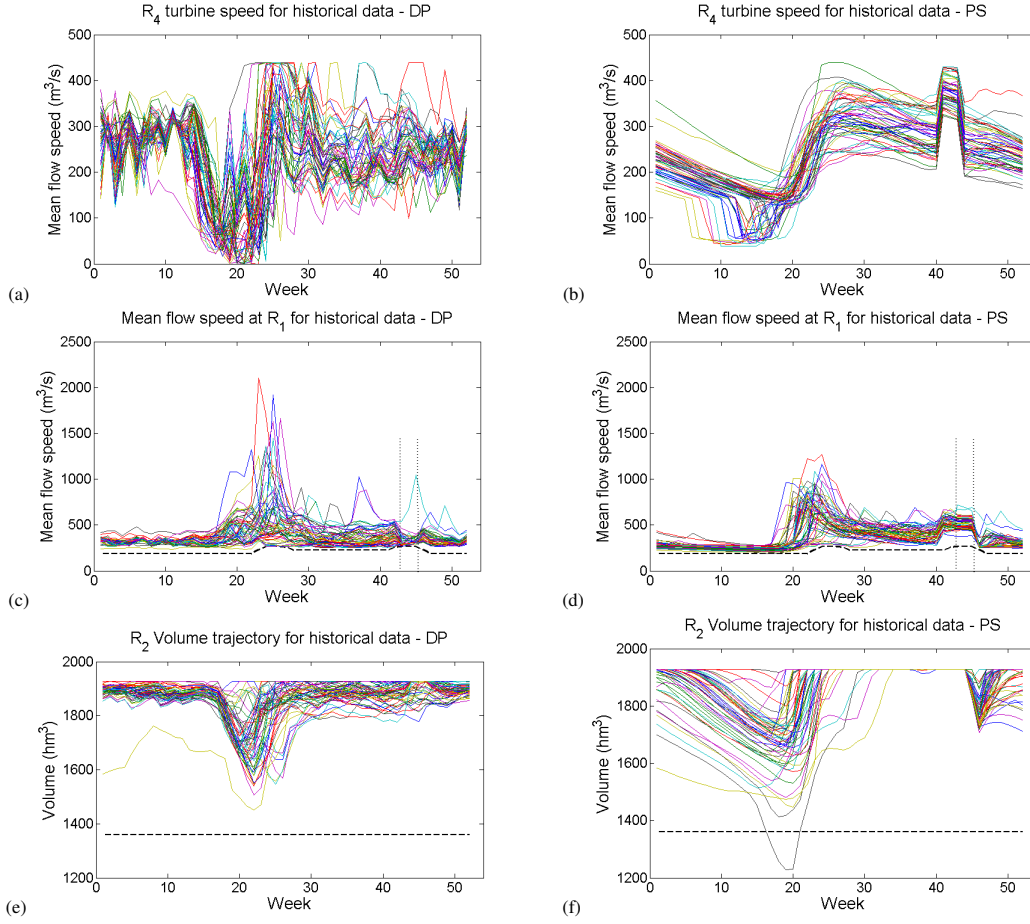

Figure 3: Qualitative comparison between *DP* and *PS with pred* solutions evaluated on the historical data. Left - *DP*, right - *PS with pred*. Plots (a)-(b) show the amount of water turbined at site $R_4$; plots (c)-(d) show the water flow at site $R_1$; plots (e)-(f) show the change in the volume of reservoir $R_2$. Dashed horizontal lines in plots (c)-(f) represent the constraints, dotted vertical lines in plots (c)-(d) mark weeks 43-45.

rithm depends on the initialization of the parameter vector, we sample the initial parameter vector uniformly at random and repeat the search 50 times. The best solution is reported.

| | *Mean-prod* | $R_1$ *v.%* | $R_1$ *43-45 v.%* | $R_1$ *43-45 v. mean* | $R_2$ *v.%* |
|---|---|---|---|---|---|
| *DP* | 8,251GW | 0% | 22% | 11 | 0% |
| *PS no pred* | 8,286GW | 0% | 28% | 2.6 | 1.8% |
| *PS with pred* | 8,290GW | 0% | 3.7% | 0.5 | 1.8% |

Table 1: Comparison between solutions found by dynamic programming (*DP*), policy search without predictive model (*PS no pred*) and policy search using the predictive model (*PS with pred*). *Mean-prod* represents the average annual electricity production; $R_1$ *v.%* is the percentage of years in which constraint 1 is violated; $R_2$ *v.%* is the percentage of years in which constraint 3 is violated; $R_1$ *43-45 v.%* is the percentage of years in which constraint 2 is violated; $R_1$ *43-45 v. mean* represents the average amount by which constraint 2 is violated.

## 5    Experimental results

We compare the solutions obtained using the proposed policy search with (*PS with pred*) and without predictive model (*PS no pred*) to a solution based on dynamic programming (*DP*), developed by Hydro-Québec. The state space of *DP* is defined by: week, water volume at each reservoir, and previous total inflow. All the continuous variables are discretized, and the transition matrix is calculated based on the PAR(1) generative model of the inflow process presented earlier. The discretization was

optimized to obtain best results. During the evaluation, the solution provided by *DP* is adjusted to avoid obviously wrong decisions, like unnecessary water spilling. All solutions are evaluated on the original historical data. The constraints in *DP* are handled in the same way as in both PS solutions, with penalties for violations taking the same form as shown previously. The only exception is the constraint 2, which requires keeping the flow roughly equal throughout several time steps. Since it is not possible to incorporate this constraint into *DP* as is, it is handled by enforcing a turbine flow between 265 $m^3/s$ (the minimum required by constraint 1) and 290 $m^3/s$.

Table 1 shows the quantitative comparison between the solutions obtained by three methods. PS solutions are able to produce more power, with the best value improving by nearly half of a percent - a sizeable improvement in the field of energy production. All solutions ensure that constraint 1 is satisfied (column $R_1$ *v.%*), but constraint 2 is more difficult. Although *PS no pred* violates this constraint slightly more often then *DP* (column $R_1$ *43-45 v.%*), the amount by which the constraint is violated is significantly smaller (column $R_1$ *43-45 v. mean*). As expected, *PS with pred* performs much better, because it explicitly incorporates inflow predictions. Finally, although both PS solutions violate constraint 3 during one out of 54 years (see Fig. 3(f)), such occasional violations are acceptable as long as they help satisfy other constraints. Overall, it is clear that *PS with pred* is a noticeable improvement over *DP* based on the quantitative comparison alone.

Practitioners are also often interested to assess the applicability of the simulated solution by other criteria that are not always captured in the problem formulation. Fig. 3 provides different plots that allow such a comparison between the *DP* and *PS with pred* solutions. Plots (a)-(b) show that the solution provided by *PS with pred* offers a significantly smoother policy compared to the *DP* solution (see also Fig. A.3 in supplementary material). This smoothness is due to the policy parametrization, while the DP roughness is the result of the discretization of the input/output spaces. Unless there are significant changes in the amount of inflows within consecutive weeks, major fluctuations in turbine speeds are undesirable, and their presence cannot be easily explained to the operator. The only fluctuations in the solution of *PS with pred* that are not the result of large inflows are cases in which the reservoir is empty (see e.g. rapid drops around 10-th week at plot (b)), or a significant increase in turbine speed around weeks 41-45 due to the change in policy parameters. This also affects the smoothness of the change in the water volume trajectory, which can be observed at plots (e)-(f) for reservoir $R_2$ for example. The period of weeks 43-45 is a reasonable exception due to the change in policy parameters that require turbining at faster speeds to satisfy constraint 2.

## 6 Discussion

We considered the problem of optimizing energy production of a hydroelectric power plant complex under several constraints. The proposed approach is based on a problem-adapted policy search whose features include predictions obtained from a predictive state representation model. The resulting solution is superior to a well-established alternative, both quantitatively and qualitatively.

It is important to point out that the proposed approach is not, in fact, specific to this problem or this domain alone. Often, real-world sequential decision problems have several decision variables, a variety of constraints of different priorities, uncertainty, etc. Incorporating all available domain knowledge into the optimization framework is often the key to obtaining acceptable solutions. This is where the policy search approach is very useful, because it is typically easy to incorporate many types of domain knowledge naturally within this framework. First, the policy space can rely on features that are deemed useful for the problem, have an interpretable structure and adhere to the constraints of the problem. Second, policy search can explore the most likely directions of improvement first, as considered by experts. Third, the policy can be evaluated directly based on its performance (regardless of the complexity of the reward function). Forth, it is usually easy to implement the policy search and parallelize parts of the policy search procedure. Finally, the use of PSRs allows us to produce good features for the policy by providing reliable predictions of future system behavior. For future work, the main objective is to evaluate the proposed approach on other realistic complex problems, in particular in domains where solutions obtained from other advanced techniques are not practically relevant.

**Acknowledgments**

We thank Grégory Emiel and Laura Fagherazzi of Hydro-Québec for many helpful discussions and for providing access to the simulator and their DP results, and Kamran Nagiyev for porting an initial version of the simulator to Java. This research was supported by the NSERC/Hydro-Québec Industrial Research Chair on the Stochastic Optimization of Electricity Generation, and by the NSERC Discovery Program.

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
