[Supplementary Material]

# Optimizing Energy Production Using Policy Search and Predictive State Representations – Appendix

## A    Complementary Plots

Figure A.1: Production functions of each site. For sites $R_2$,$R_3$ and $R_4$ the curves are given for some reference headwater of the corresponding reservoirs (shown in Fig. A.2). The largest $x$ value on each curve represents the maximum total speed of the turbines.

Figure A.2: Volume to level conversion tables for three reservoirs along with the reference points used to calculate the production curve baseline. Maximum capacities of reservoirs $R_4$, $R_3$ and $R_2$ are 3480 $hm^3$, 626.4 $hm^3$ and 1927 $hm^3$ correspondingly.

Figure A.3: Qualitative comparison between *DP* and *PS with pred* solutions evaluated on the historical data - a complementary figure to Fig. 3 in the paper. Left - *DP*, right - *PS with pred*. Plots (a)-(d) show the amount of water turbined at sites $R_3$ and $R_2$.