[Reviews · NeurIPS 2014]

Submitted by Assigned_Reviewer_11

The paper under review, "Optimizing Energy Production Using Policy Search" describes a policy search algorithm for optimizing the energy production in a hydroelectric power plant. First, the problem is specified with a model of the system, the goal and the constraints. Afterwards, a predictive state representation is introduced for the inflow process. Finally, a policy search algorithm based on a random local search is presented and evaluated on a dataset of a real power-plant. One advantages of this approach is that it can address constraints that are necessary for this application field that other methods like dynamic programming cannot directly include. Other advantages are the parallel computation framework and the easy to interpret policy representations.

The paper is clearly written and the experiments show that this approach leads to an improvement over current state of the art methods. One of the goals stated in the paper was to create an algorithm where it is easy to include domain knowledge and in my opinion the authors also developed successfully such an algorithm for a power plant system.
My main concern is that the paper is written very problem specific for a concrete hydroelectric power plant system with very special constraints. The approach contains a lot of expert information about modeling hydroelectric power plants and what constraints should be fulfilled. I think it will be difficult for readers to extract general concepts from the paper that could be applied to other application areas. If possible, an abstraction away from such a concrete system to a more general problem class would be preferable.

Some minor points:
* The naming convention for some variables is unusual. For example 'BUFFER' or 'meanR_{1}flow'.
* Provide informations about the units of the variables.
* In the paper it is mentioned that 'simulation-based optimization' is also referred as 'policy search'. But not all policy search methods are simulation-based since there also exist model-free policy search methods.
Summary: The paper "Optimizing Energy Production Using Policy Search" presents an approach based on policy search how to improve energy production in a hydroelectric power plant. The developed approach shows how to successfully combine engineering knowledge with a learning algorithm that results in a practically successful but also very problem oriented solution.

Submitted by Assigned_Reviewer_32

This is an application paper that attacks the problem of managing
turbine speeds of four dams in Quebec. The goal is to maximize energy
production (equivalent to maximizing turbine speeds) subject to
various ecological and operational constraints.

Interestingly, a full dynamical model for this problem is
available. However, this is a continuous state, continuous action,
discrete time (weekly-time-step) problem, so applying standard DP
methods to solve the problem requires discretization. This in turn
leads to non-smooth policies (i.e., turbine rates jump around because
of discretization), that are hard to explain (and undesirable for
non-represented operational reasons) to the operators.

Instead of DP, this paper pursues a PSR approach in which the
existing simulator (a custom auto-regressive model) and historical
data are combined to create a dynamical model. One relevant state
variable (month of year) is observable, so the authors incorporate
this variable into the PSR via the methods of [8]. The learn PSRs
with and without water input forecasts.

Constraints are incorporated by a mix of penalty terms, fixing the
policy in certain time periods, and partitioning the state space into
two time periods and learning a separate policy for each time period.

The results evaluate performance on many dimensions including (a)
total energy produced, (b) frequency with which constraints are
violated, and (c) severity with which constraints are violated.
(a) and (c) are improved but at a slight cost of (b). In addition, the
policies are qualitatively much nicer.

Quality: The paper is well written and the material is reasonably
self-contained, with appropriate references to prior work. The new
approach is compared to a state-of-the-art solution developed by the
Quebec water management organization, which is exactly the right
baseline. The fact that they are able to significantly improve upon
the DP policy is very interesting and has wider lessons for all of us
who work on applied reinforcement learning methods. In particular,
the need for explanable/visualizable policies is not sufficiently
appreciated in the research community. This paper helps address this
problem. This is a very nice example of applying a mix of advanced
techniques to show an important application improvement.

One question: The authors say that it is difficult (or impossible) to
incorporate the steady-flow constraint for weeks 43-45 into a DP
method. But couldn't that be done within an SMDP framework, where the
action for weeks 43-45 is treated as a single action with a 3-week
duration? This is more or less what the authors did in their
hand-coded policy for R2.

Clarity: The paper is well-written with few exceptions.

Line 142: "or" -> "of"?
insert "the" before "logarithmic case"?

Line 209: You introduce T without defining it. You need to remind
the reader what a "test" is.

Line 236: Should "assuming" be "assume"? You don't discuss Figure 2
very much. Is there more to say?

Originality: The work is original within the RL literature. I don't
know about the reservoir management literature.

Significance: wrt the research community, the paper is significant
because it shows (a) the virtues of PSRs for achieving generality, (b)
the advantages of policy search for incorporating domain constraints
and achieving smoother policies, and (c) the importance of
interpretability. In the reservoir management literature and the
larger sustainability literature, this work is important because it
demonstrates that there is significant advantage in applying the
sophisticated techniques coming out of the research community. This
paper also points the way toward many important areas for future
work. For example, under climate change, the variance in precipitation
(and hence, reservoir inputs) can be expected to increase. The
techniques in this paper could easily be applied to model the impact
of that increased variance on reservoir management and may suggest a
need for additional reservoir capacity and other management changes.

This paper is likely to be highly cited in subsequent hydropower
management work as well as in the RL community.
Summary: Excellent application paper. Lessons for the research community:
advantages of policy search for incorporating various domain
constraints. Lessons for the application community: new methods from
RL can obtain major improvements in hydropower management. Strong
potential for follow-on work.

Submitted by Assigned_Reviewer_40

This paper presents an PSR-based approach to power generation and reservoir management for a simulated hydro-electric plant. In particular, they use a PSR model to predict future upcoming inflow into the plant. They the use a policy search approach, based upon this predictive model, to develop a policy for controlling the reservoirs in the plant, and show that it improves upon a dynamic programming solution developed for this same model by Hydro-Quebec.

There are a lot of strong elements to this paper, most notably the tight integration with an application of substantial practical importance. The application of the method to hydro-electric plant is the main focal point of the paper, and as such it can serve to introduce the NIPS community to a problem that could be of substantial interest to the community.

There are also several weaknesses to the paper. Ultimately, the integration of PSRs here seems to be a relatively minor portion of the paper. In particular, the PSRs are being used here only as a means of creating a forecasting model for the upcoming flows. While this is a reasonable choice, it is by no means the only way of building such a time series model, and the paper would be much better served by considering a wide range of possible predictive models for this case. Time series forecasting for such problems is a well-studied task, and it is unclear why the authors choose to user a PSR here (as far as I can tell, effectively equivalent in this setting to subspace identification approaches for learning latent state time series models) other than this simply being a model that they were familiar with. This is probably ok, but given that this is an application paper, further discussion (and ideally implementation) of possible approaches would greatly strengthen the paper.

Given the above, it seems as though the real element that gives better performance than the DP approach is the policy search method. Again, the primary contribution here is from the applied perspective, since the policy search method itself is quite straightforward. This means that the real value here comes in the form of the particular policy and cost function chosen. This is fairly interesting from an applied perspective, but again there is little comparison to alliterative approaches. Methods like model predictive control seem like they would be a good fit here (they are ubiquitous in plant control tasks where there are hard constraints on variables), so again a discussion or evaluation of other alternative approaches seems useful.

Ultimately, the paper does seem to be along the lines of "one set of approaches tried on a problem, with reasonably good results." I don't necessarily think that this should be reason for rejecting the paper, though, as applied works can be of significant value to the community. The fact that the problem is evaluated only in simulation is also a downside, though running such control methods on real systems is highly non-trivial from a practical perspective, so this may be ok in this case. Overall, I think that this paper offers a reasonably strong applied perspective, and slightly lean toward acceptance.
Summary: This paper presents an application of predictive state representation and policy search methods to the management of a hydro-electric plant. The algorithms themselves are straightforward, and few comparisons are made to other (seemingly very reasonable) algorithmic choices, but applied aspects of the work are appealing.
Author Feedback
Author rebuttal: Thank you very much for your reviews. All the comments/suggestions will be incorporated to improve the paper.

Reviewer 11:
"My main concern is that the paper is written very problem specific for a concrete hydroelectric power plant system with very special constraints."

Indeed, the objective of the paper is to solve a specific, complex real-world problem, which means one needs to address all its specific details appropriately.
However, this problem has properties that are shared in many practical situations, such as multiple decision variables and constraints on the maximum allowed variance
of some variables spanning several time steps (which is a type of risk minimization guarantee); etc. In particular, incorporating constraints on different variables and having policies that are ``well-behaved" arise commonly in real applications.
Problems that share such characteristics could leverage our approach. We will add a discussion highlighting the main features of our problem and approach which should
be of interest in a broad range of applications (outside the context of hydro power plants).

Reviewer 32:
Regarding the constraints for weeks 43-45: using an SMDP action for the constant flow constraint is problematic, because fixing the same turbine speed for 3 consecutive weeks does not account for the inflows at site 4 and water spillage from site 3 during these weeks (which together determine the total flow at site 4). The
goal is essentially to bring the variance of the *total water flow* at site 4 within these weeks as close to 0 as possible. To implement it in an MDP setting directly,
we need to have a much finer discretization of the state space for these 3 particular weeks and to add the total flow in site 4 as a separate state variable at least
during these 3 weeks (currently, the state variable of the DP method incorporates only the total inflow to all sites added together). In addition, for week #45 the state space needs to include the total flow in site 4 that happened during week #43, since the goal is to minimize the deviation from the mean. Altogether, these state space expansions make the MDP intractable.

Reviewer 40:
"Time series forecasting for such problems is a well-studied task, and it is unclear why the authors choose to user a PSR here"
Part of the motivation for using PSRs was in fact that we wanted to test this type of model in a real application, to better understand its utility (which has not been extensively assessed in such setting). However, more importantly, we needed a model that can handle the non-Markovian inflow process. AR models are significantly
more restrictive than PSRs, hence not sufficient for this problem if the generative model is not AR by itself. As mentioned in the conclusion, we are currently investigating the sensitivity of the result of our policy search algorithm to different types of inflow models learned from the same data. These alternatives are also non-Markovian (from the class of temporal disaggregation models), are more complex and do not even have a clear way to track a sufficient statistic to predict the future. Based on our experience, PSRs seem a natural approach if the policy needs to use predictions of future inflows given a possibly complex generative model.
From a practical perspective, the core PSR learning algorithm stays the same regardless of the generative inflow model used, and hence it is easy to re-run the
algorithm with fancier generative models.

"Methods like model predictive control seem like they would be a good fit here"
We do not use model predictive control because there is a strong need in this application to produce policies that can be inspected by people, which is not natural to do in a receding-horizon setting. Linear MPC models are also not sufficient in our case, and non-linear MPC is hard to tune and prone to locally optimal solutions. Hence, the policy search approach is a more natural alternative to the dynamic programming solutions used currently in industry. We will include a discussion of these issues in the paper.